# Macro- and atomic-scale observations of a one-dimensional heterojunction in a nickel and palladium nanowire complex

Masanori Wakizaka [1✉], Shohei Kumagai [1], Hashen Wu[1], Takuya Sonobe[1], Hiroaki Iguchi[1], Takefumi Yoshida[1], Masahiro Yamashita [1,2✉] & Shinya Takaishi [1✉]

The creation of low-dimensional heterostructures for intelligent devices is a challenging research topic; however, macro- and atomic-scale connections in one-dimensional (1D) electronic systems have not been achieved yet. Herein, we synthesize a heterostructure comprising a 1D Mott insulator [Ni(chxn)$_2$Br]Br$_2$ (**1**; chxn = 1$R$-2$R$-diaminocyclohexane) and a 1D Peierls or charge-density-wave insulator [Pd(chxn)$_2$Br]Br$_2$ (**2**) using stepwise electrochemical growth. It can be considered as the first example of electrochemical liquid-phase epitaxy applied to molecular-based heterostructures with a macroscopic scale. Moreover, atomic-resolution scanning tunneling microscopy images reveal a modulation of the electronic state in the heterojunction region with a length of five metal atoms (~ 2.5 nm), that is a direct evidence for the atomic-scale connection of **1** and **2**. This is the first time that the heterojunction in the 1D chains has been shown and examined experimentally at macro- and atomic-scale. This study thus serves as proof of concept for heterojunctions in 1D electronic systems.

[1] Department of Chemistry, Graduate School of Science, Tohoku University, 6-3 Aramaki-Aza-Aoba, Aoba-Ku, Sendai 980-8578, Japan. [2] School of Materials Science and Engineering, Nankai University, Tianjin 300350, PR China. ✉email: masanori.wakizaka.a7@tohoku.ac.jp; yamasita@agnus.chem.tohoku.ac.jp; shinya.takaishi.d8@tohoku.ac.jp

Heterojunctions provide critical properties for electronic and photonic devices, e.g., charge separation and asymmetric conductance. Recently, organic–inorganic hybrids and molecule-based heterostructures have attracted great attention[1–9], as their structures and properties can be precisely tuned through crystal and molecular design. In these heterostructures, bulk[1–6] or planar heterojunctions[7–9] have been developed to obtain high performance; however, connection at the atomic scale has remained elusive so far. In 2012, Haigh et al. have synthesized and examined a heterojunction in a two-dimensional (2D) material, using graphene and hexagonal boron–nitride sheets, at the atomic scale using high-resolution transmission electron microscopy[10]. This sophisticated study demonstrated that low-dimensional heterostructures are attractive materials that show non-linear properties based on the quantum effect[11–13]. On the other hand, Fasel et al.[14] and Crommie et al.[15] reported the heterojunctions of graphene nanoribbon random copolymer at atomic-scale. However, these cases have no macro-scale heterostructures, because these heterojunctions are randomly dotted in the chains. Therefore, macro- and atomic-scale connections in one-dimensional (1D) electronic systems have not been observed yet.

Molecular nanowires may be the most promising materials to achieve 1D heterojunctions, as their electronic state and structure can be finely tuned via their molecular design. In particular, quasi-1D halogen-bridged metal complexes (MX-Chains) of the type $[M(A)_2X]Y_2$ (M = Ni, Pd, Pt; X = Cl, Br, I; A = e.g., ethylenediamine or $1R,2R$-diaminocyclohexane (chxn); Y = e.g., Cl, Br, I, or $ClO_4$) are fascinating materials that exhibit photonic[16,17], magnetic[18], and electronic properties (Fig. 1)[19–21]. MX-Chains form a specific isolated 1D electron system composed of the $d_z^2$ orbitals of the metal ions and the $p_z$ orbitals of the bridging halide ions. Many MX-Chains have been synthesized using various combinations of M, X, A, and Y[22,23]. Their electronic states and phase stability can be rationalized theoretically using the extended Peierls–Hubbard model, in which two major parameters, that is, the electron–phonon interaction ($S$) and the on-site Coulomb repulsion energies ($U$), strongly compete in such 1D electron systems[22–25]. Almost all Pd-type MX-Chains are stable in the M(II)–M(IV) mixed-valence state ($\cdots M^{II}\cdots X–M^{IV}–X\cdots$), because the stronger $S$ value ($S > U$) causes Peierls distortion. This mixed-valence state is also known as the charge-density-wave (CDW) state. In contrast, Ni-type MX-Chains invariably form

an averaged Ni(III) electronic state ($–Ni^{III}–X–$) in the Mott–Hubbard (MH) state due to the stronger $U$ value ($U > S$). Accordingly, it should be possible to develop suitable structures that involve two MX-Chains with matching crystal parameters. Previously, we have reported Ni/Pd mixed MX-Chains of the type $[Ni_{1-x}Pd_x(chxn)_2Br]Br_2$, which were obtained from the electrochemical oxidation of a mixed solution[26,27]. As the unit cell parameters for $[Ni(chxn)_2Br]Br_2$ (**1**) and $[Pd(chxn)_2Br]Br_2$ (**2**) are almost identical (Fig. 1 and Supplementary Table 1)[21,28,29], these compounds can be mixed in any ratio to produce solid solutions. Thus, combinations of these MX-Chains represent the most desirable candidate to realize a 1D heterojunction material composed of a Ni Mott insulator and a Pd Peierls insulator. Herein, we report the **1–2** heterostructure, which was synthesized via the stepwise electrochemical growth method. The 1D heterojunction was unequivocally observed using optical and scanning electron microscope energy dispersive X-ray spectroscopy (SEM-EDS) and scanning tunneling microscopy (STM), which is the first time that a macro- and atomic-scale heterojunction in a 1D electron system has been examined experimentally.

## Results

Single crystals of **1** were grown on a Pt wire electrode in a methanol (MeOH) solution of $[Ni(chxn)_2]Br_2$ via electrochemical oxidation. Subsequently, the electrodes were carefully rinsed with MeOH and then soaked in a MeOH solution of $[Pd(chxn)_2]Br_2$ containing an electrolyte. After further electrochemical oxidation, core–shell crystals[30] were obtained in which **1** is the core and **2** is the shell. After removing the crystals from the anode, the crystals were cleaved with the utmost care along the $bc$-plane using a razor blade in order to observe the **1–2** heterojunction. Figure 2a shows an optical reflection image of the **1–2** heterostructure. The domains of **1** and **2** are clearly observed as light- and dark-pink areas, respectively. On the other hand, in the SEM images, the domain of **2** has a brighter contrast than **1** (Fig. 2b and Supplementary Fig. 1). This contrast can be reasonably explained by the difference of the secondary electron (SE) yield between Pd atoms ($\delta^m$: maximum of the SE yield = 1.3) and the Ni atoms ($\delta^m = 1.0–1.3$)[31]. In addition, the SEM-EDS elemental mapping images for Ni and Pd clearly revealed the macroscopic domains corresponding to **1** and **2** (Fig. 2c, d). The opposite domains did

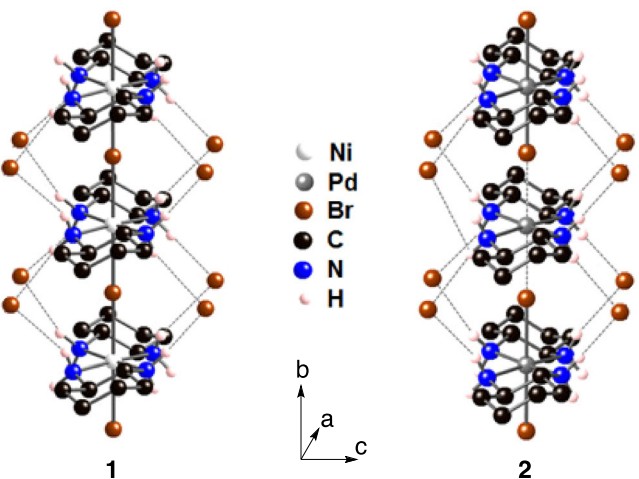

**Fig. 1 Crystal structure of the halogen-bridged metal complexes (MX-Chains).** Molecular structure of **1** and **2**[28,29]. Dashed lines represent hydrogen bonds (NH···Br). Hydrogen atoms on carbon atoms are omitted for clarity.

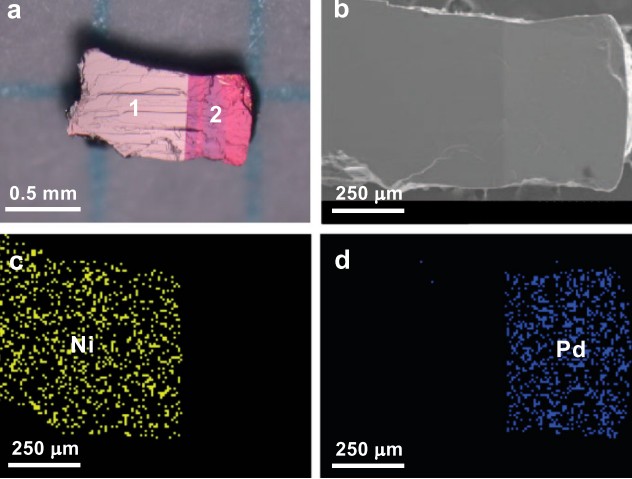

**Fig. 2 Macro-scale observation of the 1–2 heterostructure. a** Optical, **b** SEM, and **c, d** Ni and Pd EDS elemental mapping images. Scale bars, 0.5 mm and 250 μm (for **a**–**d**, respectively). Source data are provided as a Source Data file.

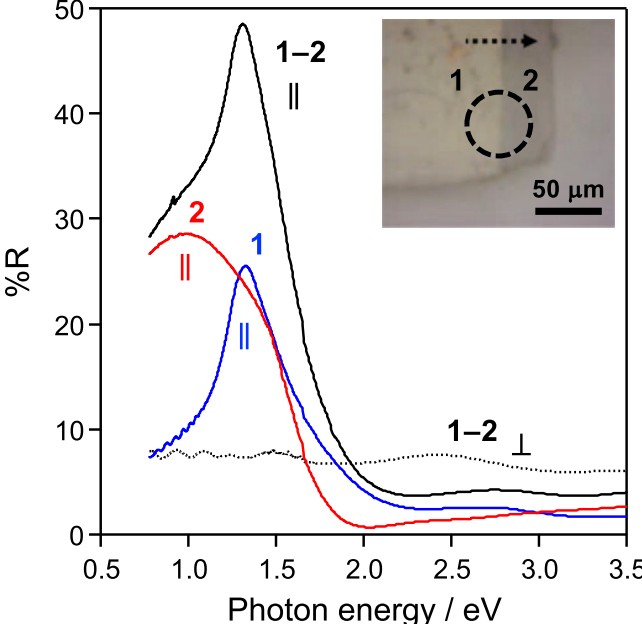

**Fig. 3 The polarized reflectivity spectra.** The spectra for parallel (∥: black solid line) and perpendicular (⊥: black dotted line) toward the chain direction of the **1–2** heterostructure. The inset shows an optical image with an aperture ($\phi = 50\,\mu m$). A scale bar 50 μm. Intensity of spectra for the **1** (blue line) and **2** (red line) regions were halved (%R×0.5). Source data are provided as a Source Data file.

not show any intensity above the noise level for Ni or Pd (Supplementary Fig. 2).

Figure 3, Supplementary Figs. 3 and 4 show polarized reflectivity spectra of the **1–2** heterostructure. The **1** and **2** regions showed sharp and broad peaks at 1.3 eV and ~ 1 eV, respectively. These bands are assigned as the charge-transfer (CT) bands (Br $4p_z$ band → upper-Hubbard-band (UHB); $Pd^{II}$ $4d_z^2$ band → $Pd^{IV}$ $4d_z^2$ band), respectively[32]. The heterojunction region showed the CT bands derived from **1** and **2** with an electric vector parallel to the chain. On the other hand, these bands entirely disappeared with an electric vector perpendicular. Therefore, the direction of 1D chains of **1** and **2** are aligned in the heterostructure.

To investigate the heterojunction at the atomic scale, STM images of the **1–2** heterostructure were recorded at room temperature by applying a positive sample bias (1.4 V) as shown in Fig. 4. The **1** region was observed as a series of oriented dots at atomic resolution (Fig. 4b, h). These dots are aligned along the $b$-axis with a spacing of ~ 5 Å (Fig. 4e), which is consistent with the Ni⋯Ni distance in the 1D chains of **1**. The **2** region shows a similar pattern of dots (Fig. 4c, i); however, the spacing along the $b$-axis is approximately double that in **1** (~ 10 Å, Fig. 4f). These spacings are intimately related to the locations of the electron-accepting orbitals in the 1D chains (Fig. 4k, l). In the MH state of **1**, the $3d_z^2$ orbitals on the Ni(III) sites can accept tunneling electrons in the unoccupied UHB, which affords one dot for every Ni(III) ion. On the other hand, in the CDW state of **2**, the electron-accepting $4d_z^2$ orbitals are only located on the Pd(IV) sites, resulting in a doubled periodicity. These patterns and their spacings are consistent with previously reported MH and CDW states in MX-Chains[23], respectively. On the other hand, the mixed structures $[Ni_{1-x}Pd_x(chxn)_2Br]Br_2$ have been reported to exhibit a phase transition from the MH to the CDW state induced by increasing Pd doping[23]. Hence, heterojunctions have not yet been observed in mixed systems due to the random substitution of the dopant metals. Figure 4d, j show an area around the

heterojunction in which **1** and **2** are clearly observed on opposite sides. Remarkably, in the central area of the heterojunction, the MH and CDW states are connected. The periodic pattern in the connected area is modulated from both the MH and CDW states. The modulation distance is estimated to be ~2.5 nm (Fig. 4g), which corresponds to five metal atoms in the chain. It should also be noted here that some defects such as steps or small islands are generated on the surface during the cleavage process. Additionally, optical and SEM images show abrupt interfaces; however, it is hardly feasible to expect that the growth edge of a Ni crystal is completely aligned at the atomic level throughout the entire crystal. The atomic-scale position of the heterojunction, i.e., the position where two kinds of chains combined, may have random distribution.

## Discussion

The mixed structures $[Ni_{1-x}Pd_x(chxn)_2Br]Br_2$ do not exhibit a connection or modulated structure shown in the **1–2** heterostructure (Fig. 4d, j, g)[23]. The modulation of the electron-accepting orbitals represents direct evidence for the connection between the MH state in **1** and the CDW state in **2** at the atomic scale. The modulations of the electronic structure and the band bending have been reported to occur at the sub-nanometer to ~2 nm scale at the heterojunction of the graphene nanoribbons[14,15], which is comparable to the modulation distance in this study (~2.5 nm). The concept of molecular-scale bandgap engineering provided by Fasel et al.[14] and Crommie et al.[15] can be applied to the MX-Chains. This work can thus be expected to advance heterojunction science by introducing a new type of molecular-scale bandgap engineering. Moreover, the novelty of this study includes not only the observation of the 1D heterojunction, but also a synthetic method for the heterostructure. As for molecular-based heterostructures such as hetero-metal-organic frameworks[33,34] and core-shell crystals[30], these heterostructures have been synthesized in a self-assembly manner by liquid-phase epitaxy (LPE). In analogy, the stepwise electrochemical growth method in this study can be considered as the first example of electrochemical LPE[35] applied to molecular-based heterostructures. The heterostructure of the MX-Chains observed in this study can be expected to significantly advance molecular-based heterostructure science. We envision that in the near future ultimate ultrasmall devices can potentially be generated that exhibit extremely narrow electron passages, i.e., single-atomic passages that are characteristic of 1D electronic systems[36-38] and thus 1D heterojunctions. The 1D heterojunction of the MX-Chains presented in this study represents a suitable model for the development of such ultrasmall devices. It should be noted here that the 1D heterojunction in this study is fundamentally different from the lateral 1D heterojunction between the side edges of 2D materials[39-41].

In conclusion, we have synthesized a molecule-based heterostructure comprising a 1D Ni Mott insulator and 1D Pd Peierls insulator via stepwise electrochemical growth. This work presents the first example of a heterojunction in a 1D electronic system at the macro- and atomic-scale and thus provides proof of concept for a heterojunction in a 1D electronic system.

## Methods

**Materials**. NiBr₂, NiBr₂·3H₂O, PdBr₂, chxn, acetone, and MeOH were purchased from Fujifilm Wako Pure Chemical Corp. Tetramethylammonium bromide (Me₄NBr) was purchased from Tokyo Chemical Industry Co. Ltd.

**Synthesis of $[Ni^{II}(chxn)_3]Br_2$**. This compound was synthesized referring to previously reported procedures[28]. Refluxing NiBr₂·3H₂O (10.0 g, 36.7 mmol) and chxn (14.6 g, 128 mmol) in MeOH (185 ml) for 2.5 h afforded a purple dispersion solution. After evaporation, acetone (40 mL) was added. A purple powder (19.8 g, 90% yield) was obtained, which was collected by filtration, washed with acetone

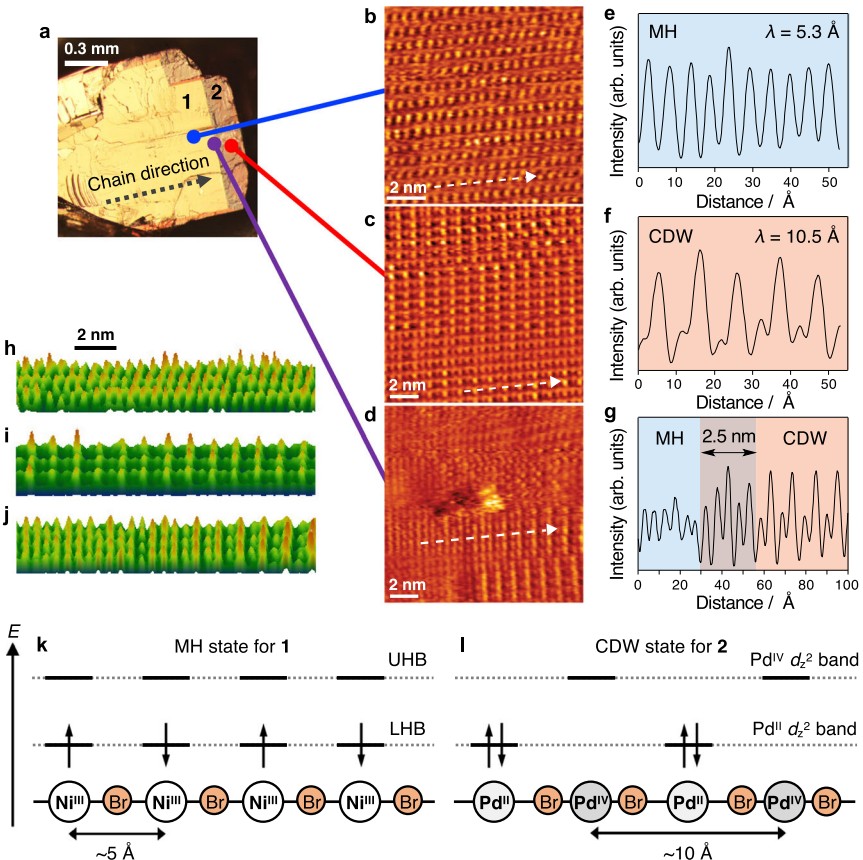

**Fig. 4 Atomic-scale observation of the 1–2 heterostructure. a** Optical polarization image of the crystal and STM current images ($V = 1.4$ V, $I = 1.0$ nA) of the **b** 1, **c** 2, and **d** heterojunction region in the **1–2** heterostructure, together with **e**, **f**, **g** the current profile in direction of the dashed arrow along the chain direction, and **h**, **i**, **j** those of 3D views. Scale bars, 0.3 mm, 2 nm and 2 nm (for **a**, **b–d**, and **h–j**, respectively). Schematic energy diagrams (**k**) of the Mott–Hubbard (MH) state for **1** and (**l**) of the charge-density-wave (CDW) state for **2**. UHB and LHB represent the upper and lower Hubbard bands, respectively. The energy levels of the bands are arbitrary, and the bands derived from the Br ions have been omitted for clarity. Source data are provided as a Source Data file.

($3 \times 20$ mL), and dried in vacuo. After dissolving 4.00 g of the obtained powder in hot MeOH (60 ml), the solution was cooled with stirring at room temperature for 4 h. A purple microcrystalline powder was obtained, which was isolated by filtration, washed with MeOH ($3 \times 5$ mL), and dried in vacuo, to afford $[Ni^{II}(chxn)_3]$ $Br_2 \cdot 2H_2O$ in 35% yield. Elemental analysis calcd (%) for $C_{18}H_{46}N_6Br_2NiO_2$ ($[Ni^{II}(chxn)_3]Br_2 \cdot 2H_2O$): C 36.21, H 7.77, N 14.07; found: C 36.29, H 7.69, N 14.20.

**Synthesis of $[Pd^{II}(chxn)_2]Br_2$.** This compound was synthesized referring to previously reported procedures[29]. Stirring $PdBr_2$ (1.00 g, 4.42 mmol) and chxn (1.52 g, 13.3 mmol) in deionized water (68 mL) for 23 h at room temperature afforded a yellow solution. After filtration and evaporation, the product was dissolved in a minimum amount of hot deionized water (~ 90 °C). Adding acetone (10–20 times amount) afforded white precipitate. A white powder was obtained, which was isolated by filtration, washed with acetone ($3 \times 10$ mL), and dried in vacuo, to afford $[Pd^{II}(chxn)_2]Br_2$ in 76% yield. Elemental analysis calcd (%) for $C_{12}H_{28}N_4Br_2Pd$ ($[Pd^{II}(chxn)_2]Br_2$): C 29.14, H 5.71, N 11.33; found: C 29.28, H 5.60, N 11.34.

**Synthesis of the 1–2 heterostructure.** Refluxing $[Ni^{II}(chxn)_3]Br_2$ (300 mg, 0.54 mmol) and $NiBr_2$ (118 mg, 0.54 mmol) in MeOH (22.5 mL) for 1 h afforded a blue solution. After the solution was cooled to room temperature, $Me_4NBr$ (800 mg, 5.2 mmol) was added as an electrolyte. An aliquot of this solution (4.0 mL) was transferred into a vial (9 mL). Then, the solution was electrolyzed at 20 μA for 3–4 days with Pt electrodes (ϕ 0.3 mm) under ambient temperature and atmosphere using a direct current multisource system (YAZAWA CS-12Z). Dark-brown single-crystals were generated on the anode. After rinsing the electrodes with MeOH, they were further electrolyzed at 20 μA for 6–8 days in a saturated MeOH (4.0 mL) solution of $[Pd^{II}(chxn)_2]Br_2$ (18 mg, 0.036 mmol) with $Me_4NBr$ (142 mg, 0.92 mmol). After rinsing the electrodes with MeOH and removing the crystals from the anode, dark-pink core–shell crystals of the **1–2** heterostructure were obtained after drying under air for several hours.

**Measurements.** SEM-EDS were recorded on a Hitachi S-4300. The polarized reflectivity spectra were recorded on a JASCO MSV-5300YMT. The reflectance (% R) was obtained from the quotient of the sample and the Al deposition mirror as a standard. STM current images were recorded using a JEOL JSPM-5200 with a Pt/Ir(20%) (ϕ 0.3 mm) probe under atmospheric pressure at room temperature. All data were summarized in an Excel file for the Source Data.

## Data availability

The data generated in this study are provided in the Source Data file. Source data are provided with this paper.

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

## Acknowledgements

This work was supported by a JSPS KAKENHI Grant Nos. JP26248015 (M.Y.), JP25810032 (S.T.), JP19H05631 (M.Y.), and JP21H01756 (M.W.) as well as the Murata Science Foundation (M21-126, M.W.). M.Y. thanks the 111 project (B18030) from China.

## Author contributions

M.W. wrote paper together with S.K. The synthesis method of the heterostructure was developed by S.K., H.W., and T.S. M.W. performed STM observation and other measurements together with S.K., H.I., T.Y., and S.T. S.T. and M.Y directed the project.

## Competing interests

The authors declare no competing interests.
