## [Peer Review File · Nature Communications]

REVIEWER COMMENTS

Reviewer #1 (Remarks to the Author):

The authors present the electrochemical synthesis of a lateral heterostructure $[\text{Ni}(\text{chxn})_2\text{Br}]\text{Br}_2$ and $[\text{Pd}(\text{chxn})_2\text{Br}]\text{Br}_2$. They characterise the different materials and the interface using scanning electron microscopy, scanning tunnelling microscopy and optical spectroscopy. The scanning tunnelling microscopy reveals the expected charge density wave modulation in the Pd material and the Mott-Hubbard state in the Ni rich material. The interface shows evidence of a mixed wave modulation. As the presence of the novel interfacial state is a key result the authors could have done more to characterise or explain this. The interface appears abrupt at the macroscale but the STM image in shows some local variations/defects at the atomic scale (Fig 4c). I also think that the method section seems a little brief to allow the synthesis to be accurately reproduced. A further minor technical query is that the authors ascribe the SEM intensity modulation to differences in atomic number while it is not stated whether the SEM image is a secondary electron or backscattered electron image. The paper is generally clear and logically presented although I felt that the justification for the importance in applications of these 1D heterostructures could have been stronger. Overall, I believe the paper has technical merit but not the depth of scientific investigation I would expect to find in a Nature Communications publication. It therefore may be better suited to a more specialised journal.

Minor typographic errors noted

In abstract – ‘prove’ should be ‘proof’

p5 ‘approximate’ should be ‘approximately’

Reviewer #2 (Remarks to the Author):

The authors report a first observation with macro-scale and atomic-scale for one-dimensional heterojunction in a nickel and palladium nanowire complex. The atomic-resolution STM images revealed a modulation of the electronic state in the heterojunction region with a length of five metal atoms (~ 2.5 nm), that is a direct evidence for the atomic-scale connection of Ni and Pd MX-chains. I am happy to recommend this manuscript for publication in Nature Communications, after minor revision.

1. If possible, please add the case of Ni and Pt MX-chain heterojunction. I like to know how the correlation length is changed.
2. If possible, please also measure IR region for polarized reflectance spectra.
3. About Fig. 2, why Ni atoms are observed in Pd region and Pd atoms are observed in Ni region in EDS maps? If the authors claim that this originates from background noise, they should adjust the cut-off level or contrast.
4. The authors should show the 2-D intensity maps for each regions of a, b, and c, from the mapping images. This is because, the author might be only showing the parts that are convenient for their discussion.
5. The authors should discuss in more detailed about a modulation of the electronic state in the heterojunction region with a length of five metal atoms (~ 2.5 nm). Why the correlation length is so short?
6. The authors should mention a perspective for possible device or application of this heterojunctions in the present 1D electronic system.

Other comments:

Abstract; prove of concept - proof of concept

Fig. 1; the size of this figure should be enlarged.

P.4; the authors should explain what the aperture is. This should be observed area of the sample under microscope attached to ultraviolet-visible spectrophotometer. The authors should show each of the real apertures for $1-2//$, $1//$, $2//$ and $1-2\perp$ on the inset figure.

Reviewer #3 (Remarks to the Author):

The authors have successfully created heterojunctions between one-dimensional Ni-Br and Pd-Br MX materials. The formation of this heterojunction is proven through scanning electron microscopy (with electron dispersive spectroscopy (SEM-EDS), reflectivity spectra, and scanning tunneling microscopy (STM). The data are of high quality, and presented clearly in the paper. Such a heterojunction has been a long standing goal in the MX community, thus this work presents a significant contribution to the MX field. However, as the work is presented the broader impact to heterostructures and associated devices is not well developed. Heterojunctions are utilized in many modern devices including solar cells, lasers, and transistors. How does this work advance

heterojunction science in general? What potential uses could these one-dimensional heterojunctions afford that are not already realized with existing structures? One strength of MX research has been the low structural dimensionality, and its use to easily model and understand phenomena on a simple scale. How might this approach help with understanding heterojunction phenomena in more complex systems and what outstanding problems might be solved with this approach? I would be willing to recommend this paper for publication in Nat Comm if the discussion were expanded to address these types of questions. At one paragraph the discussion section is quite short, and could be expanded to put the results on a broader footing that would be more appealing to the Nature readership.

Reviewer 1

We are very grateful to the referee for his/her comments. As shown below, we have addressed all comments in detail.

1 The authors present the electrochemical synthesis of a lateral heterostructure $[\text{Ni}(\text{chxn})_2\text{Br}]\text{Br}_2$ and $[\text{Pd}(\text{chxn})_2\text{Br}]\text{Br}_2$. They characterise the different materials and the interface using scanning electron microscopy, scanning tunnelling microscopy and optical spectroscopy. The scanning tunnelling microscopy reveals the expected charge density wave modulation in the Pd material and the Mott-Hubbard state in the Ni rich material. The interface shows evidence of a mixed wave modulation. As the presence of the novel interfacial state is a key result the authors could have done more to characterise or explain this. The interface appears abrupt at the macroscale but the STM image in shows some local variations/defects at the atomic scale (Fig 4c).

Thank you very much for the suggestion. The defects commented by the reviewer are related with a technical issue of STM, because STM is very sensitive to surface at one-atomic level. The crystal was cleaved before observation. As the cleavability of Ni moiety and Pd moiety is different, some defects such as steps or small islands generate on surface during the cleavage process. The atomic-scale observation was not easy; however, we succeeded as shown in Fig 4c. On the other hand, we consider that the variations commented by the reviewer are related with the crystal growth process. The heterostructure was synthesized by the stepwise electrochemical oxidations. At first, Ni crystal was synthesized on the anode, next, Pd moiety was grown on the Ni crystal. Optical and SEM images show abrupt interfaces; however, it is hardly feasible to expect that the growth edge of a Ni crystal is completely aligned at the atomic level throughout the entire crystal, which would lead to variations of the position of the heterojunction at the atomic scale. According to the reviewer's suggestion, the manuscript was revised as follows.

(Results in the original manuscript) The modulation distance is estimated to be ~ 2.5 nm, which corresponds to five metal atoms in the chain.

→

(Results in the revised manuscript) The modulation distance is estimated to be ~ 2.5 nm, which corresponds to five metal atoms in the chain. It should also be noted here that some defects such as steps or small islands are generated on the surface during the cleavage process. Additionally

optical and SEM images show abrupt interfaces; however, it is hardly feasible to expect that the growth edge of a Ni crystal is completely aligned at the atomic level throughout the entire crystal, which would lead to variations of the position of the heterojunction at the atomic scale.

2 I also think that the method section seems a little brief to allow the synthesis to be accurately reproduced.

We deeply apologize our unkind descriptions. The manuscript was revised as follows.

(Methods in the original manuscript) Refluxing $[\text{Ni}^{\text{II}}(\text{chxn})_3]\text{Br}_2$ (300 mg, 0.54 mmol) and NiBr_2 (118 mg, 0.54 mmol) in MeOH (22.5 mL) for 1 h afforded a blue solution. After the solution was cooled to room temperature, Me_4NBr (800 mg, 5.2 mmol) was added as an electrolyte. The solution was electrolyzed at 20 μA for 3–4 days with Pt electrodes (ϕ 0.3 mm) under ambient temperature and atmosphere using a direct current multisource system (YAZAWA CS-12Z). Dark-brown single-crystals were generated on the anode. After rinsing the electrodes with MeOH, they were further electrolyzed at 20 μA for 6–8 days in a saturated MeOH solution of $[\text{Pd}^{\text{II}}(\text{chxn})_2]\text{Br}_2$ with Me_4NBr . After rinsing the electrodes with MeOH and removing the crystals from the anode, dark-pink core–shell crystals of the 1–2 heterostructure were obtained after drying under air for several hours.

→

(Methods in the revised manuscript) Refluxing $[\text{Ni}^{\text{II}}(\text{chxn})_3]\text{Br}_2$ (300 mg, 0.54 mmol) and NiBr_2 (118 mg, 0.54 mmol) in MeOH (22.5 mL) for 1 h afforded a blue solution. After the solution was cooled to room temperature, Me_4NBr (800 mg, 5.2 mmol) was added as an electrolyte. An aliquot of this solution (4.0 mL) was transferred into a vial (9 mL). Then, the solution was electrolyzed at 20 μA for 3–4 days with Pt electrodes (ϕ 0.3 mm) under ambient temperature and atmosphere using a direct current multisource system (YAZAWA CS-12Z). Dark-brown single-crystals were generated on the anode. After rinsing the electrodes with MeOH, they were further electrolyzed at 20 μA for 6–8 days in a saturated MeOH (4.0 mL) solution of $[\text{Pd}^{\text{II}}(\text{chxn})_2]\text{Br}_2$ (18 mg, 0.036 mmol) with Me_4NBr (142 mg, 0.92 mmol). After rinsing the electrodes with MeOH and removing the crystals from the anode, dark-pink core–shell crystals of the 1–2 heterostructure were obtained after drying under air for several hours.

3 A further minor technical query is that the authors ascribe the SEM intensity modulation to differences in atomic number while it is not stated whether the SEM image is a secondary electron or backscattered electron image.

We deeply apologize our uncertain descriptions. The SEM observation was a secondary electron image. This contrast can be reasonably explained by the difference of the secondary electron (SE) yield between Pd atoms (δ^m : maximum of the SE yield = 1.3) and the Ni atoms ($\delta^m = 1.0-1.3$) (Seiler, H., *J. Appl. Phys.* **54**, R1–R18 (1983)).^{R1} The manuscript was revised as follows, and ref. R1 was added as ref. 31.

(Results in the original manuscript) This contrast can be reasonably explained by the different amounts of back-scattered electrons derived from the heavier Pd atoms and the lighter Ni atoms.

→

(Results in the revised manuscript) This contrast can be reasonably explained by the difference of the secondary electron (SE) yield between Pd atoms (δ^m : maximum of the SE yield = 1.3) and the Ni atoms ($\delta^m = 1.0-1.3$).³¹

(References in the original manuscript)

(30) Adolf, C. R. R., [...].

(31) Okamoto, H., [...].

(...)

→

(References in the revised manuscript)

(30) Adolf, C. R. R., [...].

(31) Seiler, H. Secondary electron emission in the scanning electron microscope, *J. Appl. Phys.* **54**, R1–R18 (1983).

(32) Okamoto, H., [...].

(...)

4 The paper is generally clear and logically presented although I felt that the justification for the importance in applications of these 1D heterostructures could have been stronger.

Thank you very much for the suggestion. Application researches seem to be attractive, but fundamentals are also important. As the previously reported graphene nanoribbon heterojunctions are random copolymers (ref. 14: Fasel, R. *et al. Nat. Nanotechnol.* **9**, 896 (2014).; ref. 15: Crommie, M. F. *et al., Nat. Nanotechnol.* **10**, 156 (2015)), the **1–2** heterostructure in this study is the first example of the heterostructure with both the 1D heterojunction at atomic- and macro-scale. Moreover, the novelty of this study includes not only the observation of the 1D heterojunction, but also a synthetic method for the heterostructure. As for molecular-based heterostructures such as hetero-metal-organic frameworks (Wöll, C. *et al. Nat. Mater.* **8**, 481 (2009);^{R2} Zhuang, J.-L. *et al. Coord. Chem. Rev.* **307**, 391 (2016)^{R3}) and core-shell crystals (ref. 30: Hosseini, M. W. *et al. J. Am. Chem. Soc.* **137**, 15390 (2015)), these heterostructures have been synthesized in a self-assembly manner by liquid-phase epitaxy (LPE). In analogy, the stepwise electrochemical growth method in this study can be considered as the first example of electrochemical LPE (ec-LPE; Maldonado, S. *et al. J. Am. Chem. Soc.* **139**, 6960 (2017))^{R4} applied to molecular-based heterostructures. The heterostructure of the MX-Chains observed in this study can be expected to significantly advance molecular-based heterostructure science.

As multi-layer heterostructures of semiconductors are very important toward applications, we are now investigating synthesis of multi-layer heterostructures of MX-Chains by ec-LPE. As a preliminary result, a triple-layer heterostructure (**1–2–1**, Ni–Pd–Ni) was achieved (Fig. R1a). The reflection spectra show that the chain directions of all layers were aligned (Figs. R1b and R1c). To understand complex systems of the multi-layer heterostructures, the simple bi-layer heterostructure in this study would be a suitable model. According to the reviewer's suggestion, the manuscript was revised as follows. Refs. R2–R4 were added as refs. 33–35.

Fig. R1. (a) Optical polarization image of a triple-layer heterostructure. (b and c) The polarized reflectivity spectra of heterojunctions (1–2 and 2–1) for parallel (||: black solid line) and perpendicular (⊥: black dotted line) toward the chain direction of the 1–2–1 heterostructure. The inset shows an optical image with an aperture ($\phi = 100 \mu\text{m}$). Intensity of spectra for the 1 (blue line) and 2 (red line) regions were halved ($\%R \times 0.5$).

(Abstract in the original manuscript) We have synthesized a heterostructure comprising a one-dimensional (1D) Mott insulator $[\text{Ni}^{\text{III}}(\text{chxn})_2\text{Br}]\text{Br}_2$ (**1**; chxn = 1*R*-2*R*-diaminocyclohexane) and a 1D Peierls or charge-density-wave insulator $[\text{Pd}^{\text{II}}(\text{chxn})_2][\text{Pd}^{\text{IV}}(\text{chxn})_2\text{Br}_2]\text{Br}_4$ (**2**; hereafter abbreviated as $[\text{Pd}(\text{chxn})_2\text{Br}]\text{Br}_2$) using stepwise electrochemical growth. The macroscopic heterostructure was confirmed by optical and scanning electron microscopy (SEM).

→

(Abstract in the revised manuscript) We have synthesized a heterostructure comprising a one-dimensional (1D) Mott insulator $[\text{Ni}^{\text{III}}(\text{chxn})_2\text{Br}]\text{Br}_2$ (**1**; chxn = 1*R*-2*R*-diaminocyclohexane) and a 1D Peierls or charge-density-wave insulator $[\text{Pd}^{\text{II}}(\text{chxn})_2][\text{Pd}^{\text{IV}}(\text{chxn})_2\text{Br}_2]\text{Br}_4$ (**2**; hereafter abbreviated as $[\text{Pd}(\text{chxn})_2\text{Br}]\text{Br}_2$) using stepwise electrochemical growth. **It can be considered as the first example of electrochemical liquid-phase epitaxy (ec-LPE) applied to molecular-based heterostructures.** The macroscopic heterostructure was confirmed by optical and scanning electron microscopy (SEM).

(Discussion in the original manuscript) The mixed structures $[\text{Ni}_{1-x}\text{Pd}_x(\text{chxn})_2\text{Br}]\text{Br}_2$ do not exhibit such a connection or modulated structure shown in the **1–2** heterostructure (Fig. 4c).²³ The modulation of the electron-accepting orbitals is a direct evidence for the connection between the MH state in **1** and the CDW state in **2** at the atomic scale.

→

(Discussion in the revised manuscript) The mixed structures $[\text{Ni}_{1-x}\text{Pd}_x(\text{chxn})_2\text{Br}]\text{Br}_2$ do not exhibit a connection or modulated structure shown in the **1–2** heterostructure (Fig. 4c).²³ The modulation of the electron-accepting orbitals represents direct evidence for the connection between the MH state in **1** and the CDW state in **2** at the atomic scale. [...]. Moreover, the novelty of this study includes not only the observation of the 1D heterojunction, but also a synthetic method for the heterostructure. As for molecular-based heterostructures such as hetero-metal-organic frameworks^{33,34} and core-shell crystals,³⁰ these heterostructures have been synthesized in a self-assembly manner by liquid-phase epitaxy (LPE). In analogy, the stepwise electrochemical growth method in this study can be considered as the first example of electrochemical LPE (ec-LPE)³⁵ applied to molecular-based heterostructures. The heterostructure of the MX-Chains observed in this study can be expected to significantly advance molecular-based heterostructure science.

(References in the original manuscript)

(31) Okamoto, H., [...].

(...)

→

(References in the revised manuscript)

(32) Okamoto, H., [...].

(33) Shekhah, O., Wang, H., Paradinas, M., Ocal, C., Schüpbach, B., Terfort, A., Zacher, D., Fischer, R. A. & Wöll, C. Controlling interpenetration in metal–organic frameworks by liquid-phase epitaxy, *Nat. Mater.* **8**, 481–484 (2009).

(34) Zhuang, J.-L., Terfort, A. & Wöll, C. Formation of oriented and patterned films of metal–organic frameworks by liquid phase epitaxy: a review, *Coord. Chem. Rev.* **307**, 391–424 (2016).

(35) Demuth, J., Fahrenkrug, E., Ma, L., Shodiya, T., Deitz, J. I., Grassman, T. J. & Maldonado, S. Electrochemical liquid phase epitaxy (ec-LPE): a new methodology for the synthesis of crystalline group iv semiconductor epilayers, *J. Am. Chem. Soc.* **139**, 6960–6968 (2017).

(...)

5 Overall, I believe the paper has technical merit but not the depth of scientific investigation I would expect to find in a Nature Communications publication. It therefore may be better suited to a more specialised journal.

Thank you very much for the suggestions. This work demonstrated for the first time the heterostructure with 1D electronic system and revealed that two-kinds of molecular-based chains are connected at the 1D heterojunction. These results can be expected to significantly advance molecular-based heterostructure science and should be valuable above a technical merit. We believe that the revised manuscript satisfies the reviewer's requirements.

Minor typographic errors noted

In abstract – 'prove' should be 'proof'

p5 'approximate' should be 'approximately'

Thank you very much for pointing out, and we apologize our typographic errors. These were corrected as follows.

(Results in the original manuscript) The **2** region shows a similar pattern of dots (Fig. 4b); however, the spacing along the *b*-axis is approximate double that in **1** (~10 Å).

→

(Results in the revised manuscript) The **2** region shows a similar pattern of dots (Fig. 4b); however, the spacing along the *b*-axis is approximately double that in **1** (~10 Å).

(Abstract in the original manuscript) This study thus serves as prove of concept for heterojunctions in 1D electronic systems.

→

(Abstract in the revised manuscript) This study thus serves as proof of concept for heterojunctions in 1D electronic systems.

Reviewer 2

We are very grateful to the referee for his/her comments. As shown below, we have addressed all comments in detail.

The authors report a first observation with macro-scale and atomic-scale for one-dimensional heterojunction in a nickel and palladium nanowire complex. The atomic-resolution STM images revealed a modulation of the electronic state in the heterojunction region with a length of five metal atoms (~2.5 nm), that is a direct evidence for the atomic-scale connection of Ni and Pd MX-chains. I am happy to recommend this manuscript for publication in Nature Communications, after minor revision.

Thank you very much for understanding importance of this work.

1. If possible, please add the case of Ni and Pt MX-chain heterojunction. I like to know how the correlation length is changed.

Thank you very much for the suggestion. There are no reports for the electrochemical synthesis of Pt MX-chains ever. Pt MX-chains have been synthesized by the chemical oxidation methods using Cl_2 , Br_2 , I_2 , or H_2O_2 . Thus, we tried the electrochemical synthesis as shown in Fig. R2. The Cl-bridged compound could not be obtained. The Br-bridged compound was obtained only as polycrystalline solids, which is not suitable for synthesis of the heterostructures. On the other hand, the I-bridged compound was obtained as single crystals on the anode. It can be suitable for synthesis of the heterostructures. However, there are no reports for the I-bridged compounds of Ni and Pd MX-chains ever. These compounds are considered to be unstable relating with redox potentials between Ni/Pd and I ions. Therefore, the heterostructure using Pt MX-chains cannot be synthesized at present.

Fig. R2. The electrochemical synthesis for Pt-MX chains.

2. If possible, please also measure IR region for polarized reflectance spectra.

Thank you very much for the suggestion. We measured IR and UV-Vis-NIR spectra for polarized reflectance as shown in Figs. R3 (Ni moiety of far place from the heterojunction), R4 (Ni moiety of near place from the heterojunction), R5 (the heterojunction area), R6 (Pd moiety), and R7 (a single crystal of $[\text{Ni}(\text{chxn})_2\text{Br}]\text{Br}_2$). IR spectra were recorded on a JASCO IRT-7000, whereas UV-Vis-NIR spectra were recorded on a JASCO MSV-5300YMT. Interestingly, Ni moieties of the heterostructure showed a broad peak in IR region (~ 0.3 eV). On the other hand, Pd moiety of the heterostructure did not show such peak. However, a single crystal of $[\text{Ni}(\text{chxn})_2\text{Br}]\text{Br}_2$ showed the corresponding peak in IR region. Therefore, we considered that the broad peak in IR region is derived from $[\text{Ni}(\text{chxn})_2\text{Br}]\text{Br}_2$ itself rather than the heterojunction. The reflection spectra for $[\text{M}(\text{chxn})_2\text{Br}]\text{Br}_2$ ($\text{M} = \text{Ni}, \text{Pd}, \text{Pt}$) were reported in 1990 (Fig. R8; ref. 32. (original ref. 31): Okamoto, H. *et al. Phys. Rev. B* **42**, 10381 (1990)), but it reported until ~ 0.5 eV. The broad peak in IR region has not reported ever and its origin is unknown, thus, we are investigating as the future work.

Fig. R3. (a) IR and (b) UV-Vis-NIR with IR spectra of the Ni moiety (far place from the heterojunction) for parallel and perpendicular toward the chain direction of the **1–2** heterostructure. The insets show optical images with apertures ($100 \text{ nm} \times 100 \text{ nm}$, $\phi = 100 \mu\text{m}$).

Fig. R4. (a) IR and (b) UV-Vis-NIR with IR spectra of the Ni moiety (near place from the heterojunction) for parallel and perpendicular toward the chain direction of the **1–2** heterostructure. The insets show optical images with apertures ($100 \text{ nm} \times 100 \text{ nm}$, $\phi = 100 \mu\text{m}$).

Fig. R5. (a) IR and (b) UV-Vis-NIR with IR spectra of the heterojunction for parallel and perpendicular toward the chain direction of the 1–2 heterostructure. The insets show optical images with apertures ($100 \text{ nm} \times 100 \text{ nm}$, $\phi = 100 \mu\text{m}$).

Fig. R6. (a) IR and (b) UV-Vis-NIR with IR spectra of the Pd moiety for parallel and perpendicular toward the chain direction of the 1–2 heterostructure. The insets show optical images with apertures ($100 \text{ nm} \times 100 \text{ nm}$, $\phi = 100 \mu\text{m}$).

Fig. R7. (a) IR and (b) UV-Vis-NIR with IR spectra of $[\text{Ni}(\text{chxn})_2\text{Br}]\text{Br}_2$ for parallel and perpendicular toward the chain direction. The insets show optical images with apertures ($100 \text{ nm} \times 100 \text{ nm}$, $\phi = 100 \mu\text{m}$).

Fig. R8. Reflection spectra for $[\text{M}(\text{chxn})_2\text{Br}]\text{Br}_2$ ($\text{M} = \text{Ni}, \text{Pd}, \text{Pt}$) in ref. 32. (original ref. 31): Okamoto, H. *et al. Phys. Rev. B* **42**, 10381 (1990).

3. About Fig. 2, why Ni atoms are observed in Pd region and Pd atoms are observed in Ni region in EDS maps? If the authors claim that this originates from background noise, they should adjust the cut-off level or contrast.

Thank you very much for the suggestion. Ni atoms are not observed in Pd region and vice versa. It is noise. The opposite domains did not show any intensity above the noise level for Ni or Pd (Supplementary Fig. 2). According to the reviewer's suggestion, the manuscript and Fig. 2 were revised as follows.

(Results in the original manuscript) In addition, the SEM-EDS elemental mapping images for Ni and Pd clearly revealed the macroscopic domains corresponding to **1** and **2** (Figs. 2c and 2d, and Supplementary Fig. 2).

→

(Results in the revised manuscript) In addition, the SEM-EDS elemental mapping images for Ni and Pd clearly revealed the macroscopic domains corresponding to **1** and **2** (Figs. 2c and 2d). The opposite domains did not show any intensity above the noise level for Ni or Pd (Supplementary Fig. 2).

(Fig. 2 in the original manuscript)

→

(Fig. 2 in the revised manuscript)

4. The authors should show the 2-D intensity maps for each regions of a, b, and c, from the mapping images. This is because, the author might be only showing the parts that are convenient for their discussion.

Thank you very much for the suggestion. The 2-D intensity maps for a, b, and c were added in Fig. 4 as a', b', and c', respectively. The current profile of Fig. 4a, b, and c is a single line, whereas these 2-D intensity maps show multiple chains, which supports our discussion. On the other hand, it is very difficult to obtain a larger area for 2-D intensity map of the heterojunction because of a technical issue of STM. The crystal was cleaved before observation. As the cleavability of Ni moiety and Pd moiety is different, some defects such as steps or small islands generate on surface during the cleavage process. The atomic-scale observation was not easy; however, we succeeded as shown in Fig 4c. Additionally, the heterostructure was synthesized by the stepwise electrochemical oxidations. At first, Ni crystal was synthesized on the anode, next, Pd moiety was grown on the Ni crystal. Optical and SEM images show abrupt interfaces; however, it is hardly feasible to expect that the growth edge of a Ni crystal is completely aligned at the atomic level throughout the entire crystal, which would lead to variations of the position of the heterojunction at the atomic scale. According to the reviewer's suggestion, the manuscript and Fig. 4 was revised as follows.

(Results in the original manuscript) The modulation distance is estimated to be ~ 2.5 nm, which corresponds to five metal atoms in the chain.

→

(Results in the revised manuscript) The modulation distance is estimated to be ~ 2.5 nm, which

corresponds to five metal atoms in the chain. It should also be noted here that some defects such as steps or small islands are generated on the surface during the cleavage process. Additionally optical and SEM images show abrupt interfaces; however, it is hardly feasible to expect that the growth edge of a Ni crystal is completely aligned at the atomic level throughout the entire crystal, which would lead to variations of the position of the heterojunction at the atomic scale.

(Fig. 4 in the original manuscript)

Fig. 4 | Atomic-scale observation of the 1–2 heterostructure. Optical polarization image of the crystal and STM current images ($V = 1.4 \text{ V}$, $I = 1.0 \text{ nA}$) of the (a) 1, (b) 2, and (c) heterojunction region in the 1–2 heterostructure, together with the current profile in direction of the dashed arrow along the chain direction. (d) Schematic energy diagrams of the MH state for 1 and the CDW state for 2. UHB and LHB represent the upper and lower Hubbard bands, respectively. The energy levels of the bands are arbitrary, and the bands derived from the Br ions have been omitted for clarity.

→

(Fig. 4 in the revised manuscript)

Fig. 4 | Atomic-scale observation of the 1–2 heterostructure. Optical polarization image of the crystal and STM current images ($V = 1.4$ V, $I = 1.0$ nA) of the (a) **1**, (b) **2**, and (c) heterojunction region in the 1–2 heterostructure, together with the current profile in direction of the dashed arrow along the chain direction, and (a', b', c') those of 3D views. (d) Schematic energy diagrams of the MH state for **1** and the CDW state for **2**. UHB and LHB represent the upper and lower Hubbard bands, respectively. The energy levels of the bands are arbitrary, and the bands derived from the Br ions have been omitted for clarity.

5. The authors should discuss in more detailed about a modulation of the electronic state in the heterojunction region with a length of five metal atoms (~2.5 nm). Why the correlation length is so short?

Thank you very much for the suggestion, and we deeply apologize our lack of explanation in the manuscript. The modulations of the electronic structure and the band bending were reported to occur at sub-nanometer to ~2 nm scale at the heterojunction of the graphene nanoribbons (ref. 14: Fasel, R. *et al. Nat. Nanotechnol.* **9**, 896–900 (2014); ref. 15: Crommie, M. F. *et al. Nat. Nanotechnol.* **10**, 156–160 (2015)), which is comparable to the modulation distance in this study (~2.5 nm). Therefore, it can be suitable length of 1D heterojunctions. The manuscript was revised as follows.

(Discussion in the original manuscript) The modulation of the electron-accepting orbitals is a direct evidence for the connection between the MH state in **1** and the CDW state in **2** at the atomic scale.

→

(Discussion in the revised manuscript) The modulation of the electron-accepting orbitals **represents** direct evidence for the connection between the MH state in **1** and the CDW state in **2** at the atomic scale. **The modulations of the electronic structure and the band bending have been reported to occur at the sub-nanometer to ~2 nm scale at the heterojunction of the graphene nanoribbons,^{14,15} which is comparable to the modulation distance in this study (~2.5 nm).**

6. The authors should mention a perspective for possible device or application of this heterojunctions in the present 1D electronic system.

Thank you very much for the suggestion. In the near future, ultimate ultrasmall devices can potentially be generated that exhibit extremely narrow electron passages, i.e., single-atomic passages that are characteristic of 1D electronic systems and thus 1D heterojunctions. The 1D heterojunction of the MX-Chains presented in this study represents a suitable model for the development of such ultrasmall devices. The manuscript was revised as follows.

(Discussion in the original manuscript) It should be noted here that the 1D heterojunction in this study is fundamentally different from the lateral 1D heterojunction between the side edges of 2D materials.^{32–34}

→

(Discussion in the revised manuscript) In the near future, ultimate ultrasmall devices can potentially be generated that exhibit extremely narrow electron passages, i.e., single-atomic passages that are characteristic of 1D electronic systems and thus 1D heterojunctions. The 1D heterojunction of the MX-Chains presented in this study represents a suitable model for the development of such ultrasmall devices. It should be noted here that the 1D heterojunction in this study is fundamentally different from the lateral 1D heterojunction between the side edges of 2D materials.³⁶⁻³⁸

Other comments:

Abstract; prove of concept - proof of concept

Fig. 1; the size of this figure should be enlarged.

Thank you very much for pointing out, and we apologize our typographic error and the small figure. The typographic error and Fig. 1 were corrected as follows.

(Abstract in the original manuscript) This study thus serves as prove of concept for heterojunctions in 1D electronic systems.

→

(Abstract in the revised manuscript) This study thus serves as **proof** of concept for heterojunctions in 1D electronic systems.

(Fig. 1 in the original manuscript)

→

(Fig. 1 in the revised manuscript)

P.4; the authors should explain what the aperture is. This should be observed area of the sample under microscope attached to ultraviolet-visible spectrophotometer. The authors should show each of the real apertures for 1-2//, 1//, 2// and 1-2 \perp on the inset figure.

Thank you very much for the suggestion. The apertures were used $\phi = 50 \mu\text{m}$ for **1-2** (//, \perp) and **1** (//, \perp), and $10 \mu\text{m} \times 100 \mu\text{m}$ for **2** (//, \perp). These apertures were highlighted by the dotted circle or square lines and have been shown in Fig. 3 and Supplementary Fig. 3. The spectra for // and \perp were recorded using the same apertures at the same positions. The real apertures were shown in Fig. R9, but they are hard to see. Thus, the apertures were highlighted.

Fig. R9. The real apertures of the reflection spectra for (a) **1-2** (// and \perp , $\phi = 50 \mu\text{m}$), (b) **1** (// and \perp , $\phi = 50 \mu\text{m}$), and (c) **2** (// and \perp , $10 \mu\text{m} \times 100 \mu\text{m}$).

Reviewer 3

We are very grateful to the referee for his/her comments. As shown below, we have addressed all comments in detail.

The authors have successfully created heterojunctions between one-dimensional Ni-Br and Pd-Br MX materials. The formation of this heterojunction is proven through scanning electron microscopy (with electron dispersive spectroscopy (SEM-EDS), reflectivity spectra, and scanning tunneling microscopy (STM). The data are of high quality, and presented clearly in the paper. Such a heterojunction has been a long standing goal in the MX community, thus this work presents a significant contribution to the MX field.

Thank you very much for understanding importance of this work.

However, as the work is presented the broader impact to heterostructures and associated devices is not well developed. Heterojunctions are utilized in many modern devices including solar cells, lasers, and transistors. How does this work advance heterojunction science in general?

Thank you very much for the suggestion. The modulations of the electronic structure and the band bending have been reported to occur at the sub-nanometer to ~ 2 nm scale at the heterojunction of the graphene nanoribbons (ref. 14: Fasel, R. *et al. Nat. Nanotechnol.* **9**, 896–900 (2014); ref. 15: Crommie, M. F. *et al. Nat. Nanotechnol.* **10**, 156–160 (2015)), which is comparable to the modulation distance in this study (~ 2.5 nm). The concept of molecular-scale bandgap engineering provided by Fasel *et al.*¹⁴ and Crommie *et al.*¹⁵ can be applied to the MX-Chains. This work can thus be expected to advance heterojunction science by introducing a new type of molecular-scale bandgap engineering. The manuscript was revised as follows.

(Discussion in the original manuscript) The modulation of the electron-accepting orbitals is a direct evidence for the connection between the MH state in **1** and the CDW state in **2** at the atomic scale.

→

(Discussion in the revised manuscript) The modulation of the electron-accepting orbitals **represents** direct evidence for the connection between the MH state in **1** and the CDW state in **2** at the atomic scale. **The modulations of the electronic structure and the band bending have been reported to occur at the sub-nanometer to ~ 2 nm scale at the heterojunction of the graphene**

nanoribbons,^{14,15} which is comparable to the modulation distance in this study (~2.5 nm). The concept of molecular-scale bandgap engineering provided by Fasel *et al.*¹⁴ and Crommie *et al.*¹⁵ can be applied to the MX-Chains. This work can thus be expected to advance heterojunction science by introducing a new type of molecular-scale bandgap engineering.

What potential uses could these one-dimensional heterojunctions afford that are not already realized with existing structures?

Thank you very much for the suggestion. In the near future, ultimate ultrasmall devices can potentially be generated that exhibit extremely narrow electron passages, i.e., single-atomic passages that are characteristic of 1D electronic systems and thus 1D heterojunctions. They have not been achieved yet. The 1D heterojunction of the MX-Chains presented in this study represents a suitable model for the development of such ultrasmall devices. The manuscript was revised as follows.

(Discussion in the original manuscript) It should be noted here that the 1D heterojunction in this study is fundamentally different from the lateral 1D heterojunction between the side edges of 2D materials.³²⁻³⁴

→

(Discussion in the revised manuscript) In the near future, ultimate ultrasmall devices can potentially be generated that exhibit extremely narrow electron passages, i.e., single-atomic passages that are characteristic of 1D electronic systems and thus 1D heterojunctions. The 1D heterojunction of the MX-Chains presented in this study represents a suitable model for the development of such ultrasmall devices. It should be noted here that the 1D heterojunction in this study is fundamentally different from the lateral 1D heterojunction between the side edges of 2D materials.³⁶⁻³⁸

One strength of MX research has been the low structural dimensionality, and its use to easily model and understand phenomena on a simple scale. How might this approach help with understanding heterojunction phenomena in more complex systems and what outstanding problems might be solved with this approach?

Thank you very much for the suggestion. We consider that Tomonaga–Luttinger liquid (TLL) would be an outstanding problem of the 1D heterojunctions. TLL is considered as a unique phenomenon in 1D electronic system and considers strong electron repulsion which does not

appear in Fermi liquid. TLL has might been considered as theoretical interest in early stage (Tomonaga, S., *Prog. Theor. Phys.* **5**, 544–569 (1950); Luttinger, J. M., *J. Math. Phys.* **4**, 1154–1162 (1963); Mattis, D. C. *et al.*, *J. Math. Phys.* **6**, 304–312 (1965)); however, it was demonstrated experimentally in recent years (Bockrath, M. *et al.* *Nature* **397**, 598–601 (1999); Yao, Z. *et al.* *Nature* **402**, 273–276 (1999); Ishii, H., *et al.*, *Nature* **426**, 540–544 (2003)). We are under investigation heterojunction phenomena of TLL in the MX-Chains.

I would be willing to recommend this paper for publication in Nat Comm if the discussion were expanded to address these types of questions. At one paragraph the discussion section is quite short, and could be expanded to put the results on a broader footing that would be more appealing to the Nature readership.

Thank you very much for the suggestions, and we believe that the revised manuscript satisfies the reviewer's requirements.

Other revisions

(Acknowledgment in the original manuscript) This work was supported by a JSPS KAKENHI Grant Nos. 26248015, 25810032, and 19H05631.

→

(Acknowledgment in the revised manuscript) This work was supported by a JSPS KAKENHI Grant Nos. 26248015, 25810032, 19H05631, and 21H01756 as well as the Murata Science Foundation (M21-126).

(Contributions in the original manuscript) M.W. performed STM observation together with H.I., T.Y., and S.T.

→

(Contributions in the revised manuscript) M.W. performed STM observation and other measurements together with S.K., H.I., T.Y., and S.T.

REVIEWERS' COMMENTS

Reviewer #1 (Remarks to the Author):

In their response the authors have provided possible explanation for the variability of the interface junction at the nano/atomic scale so that this discussion now better represents the data. Unfortunately they have not provided any additional characterisation of the magnitude of this nanoscale variability. It would have been helpful for the authors to have quantified this and also to have commented on the likely potential effect this will have on the other measurements and proposed applications. Nonetheless I feel the paper now clearly represents the results obtained.

The authors have made some effort to discuss the application of the MX synthesis and place the work in the broader context of the field as requested all reviewers. However, I feel their mention of "1D electronics systems" and "ultrasmall devices" is still a bit vague in this regard.

Minor typographical errors have been corrected, the methods expanded and the presentation of figures improved where necessary.

Reviewer #2 (Remarks to the Author):

I am now satisfied with the revised manuscript for publication in Nature Communications. Please include the IR region for polarized reflectance spectra in SI.

Reviewer #3 (Remarks to the Author):

I recommend publishing the revised manuscript as a Nature Communication. The authors have addressed my concerns with the original manuscript.

Response for the Reviewer 1

We are very grateful to the referee for his/her comments. As shown below, we have addressed all comments in detail.

In their response the authors have provided possible explanation for the variability of the interface junction at the nano/atomic scale so that this discussion now better represents the data. Unfortunately they have not provided any additional characterisation of the magnitude of this nanoscale variability. It would have been helpful for the authors to have quantified this and also to have commented on the likely potential effect this will have on the other measurements and proposed applications. Nonetheless I feel the paper now clearly represents the results obtained.

Thank you very much for the suggestive comments. The sentence, “Additionally optical and SEM images show abrupt interfaces; however, it is hardly feasible to expect that the growth edge of a Ni crystal is completely aligned at the atomic level throughout the entire crystal, which would lead to variations of the position of the heterojunction at the atomic scale.”, was added in the previous revision. The reviewer pointed out here about “the variability of the interface junction”. We carefully considered the reviewer’s comments, and we wonder that the reviewer has been expected about the variability of modulation property of the heterojunction. If so, we deeply apologize for our unkind description. We did not mean to discuss about the variability of modulation property of the heterojunction. We meant to describe about the positions of the heterojunction, i.e. the position where two kinds of chains combined. The atomic-scale position of the heterojunction may have random distribution, because it is hardly feasible to expect that the growth edge of a Ni crystal is completely aligned at the atomic level throughout the entire crystal. In Figure 4c (the revised Figure 4d), several lines of MX-Chains show the heterojunction at the same position, but other lines (e.g. upper lines) don’t show it, which can be consistent with the idea of distribution of positions of the heterojunction. On the other hand, we are very glad that the reviewer feels the paper now clearly represents the results obtained. We would like to revise the sentence as follows, and we wish our response dispels the reviewer’s concern. If there are any further comments, we are pleased to response.

(Results in the original manuscript)

Additionally optical and SEM images show abrupt interfaces; however, it is hardly feasible to expect that the growth edge of a Ni crystal is completely aligned at the atomic level throughout the entire crystal, which would lead to variations of the position of the heterojunction at the

atomic scale.

→

(Results in the revised manuscript)

Additionally, optical and SEM images show abrupt interfaces; however, it is hardly feasible to expect that the growth edge of a Ni crystal is completely aligned at the atomic level throughout the entire crystal. The atomic-scale position of the heterojunction, i.e. the position where two kinds of chains combined, may have random distribution.

The authors have made some effort to discuss the application of the MX synthesis and place the work in the broader context of the field as requested all reviewers. However, I feel their mention of "1D electronics systems" and "ultrasmall devices" is still a bit vague in this regard.

Thank you very much for the suggestion. In the previous revision, the reviewer 3 commented “*What potential uses could these one-dimensional heterojunctions afford that are not already realized with existing structures?*”. Our response was “In the near future, ultimate ultrasmall devices can potentially be generated that exhibit extremely narrow electron passages, i.e., single-atomic passages that are characteristic of 1D electronic systems and thus 1D heterojunctions. They have not been achieved yet. The 1D heterojunction of the MX-Chains presented in this study represents a suitable model for the development of such ultrasmall devices.” Then, the reviewer 3 accepted. On the other hand, the reviewer 1 here commented "ultrasmall devices" is still a bit vague. This part cannot be made a big change, as this part is the response for the reviewer 1’s question and has been accepted. We carefully considered and felt that the comment here is reasonable, because "ultrasmall devices" is not realized to date. It is a bit vague idea. Therefore, the sentence was revised as follows. In addition, the references that attempt to make narrow electron passages of the 1D electronic systems were added (Mas-Ballesté *et al.*, *Chem. Soc. Rev.* **39**, 4220–4233 (2010); Welte *et al.*, *Nat. Nanotechnol.* **5**, 110–115 (2010); Hermosa *et al.*, *Nat. Commun.* **4**, 1709 (2013)). We wish these revisions dispel the reviewer’s concern. It is noted that "1D electronics systems" is not contained in our paper. “Electronics” and “electronic system” is different terminology. If there are any further comments, we are pleased to response.

(Discussion in the original manuscript)

In the near future, ultimate ultrasmall devices can potentially be generated that exhibit extremely narrow electron passages, i.e., single-atomic passages that are characteristic of 1D electronic systems and thus 1D heterojunctions.

→

(Discussion in the revised manuscript)

In the near future, although it is a bit vague idea, ultimate ultrasmall devices can potentially be generated that exhibit extremely narrow electron passages, i.e., single-atomic passages that are characteristic of 1D electronic systems^{36–38} and thus 1D heterojunctions.

(References in the original manuscript)

[...]

(35) Demuth, J., [...].

(36) Li, M.-Y., [...].

[...]

→

(References in the revised manuscript)

(35) Demuth, J., [...].

(36) Mas-Ballesté, R., Gómez-Herrero, J. & Zamora, F. One-dimensional coordination polymers on surfaces: towards single molecule devices, *Chem. Soc. Rev.* **39**, 4220–4233 (2010).

(37) Welte, L., Calzolari, A., Felice, R. D., Zamora, F. & Gómez-Herrero, J. Highly conductive self-assembled nanoribbons of coordination polymers, *Nat. Nanotechnol.* **5**, 110–115 (2010).

(38) Hermosa, C., Álvarez, J. V., Azani, M.-R., Gómez-García, C. J., Fritz, M., Soler, J. M., Gómez-Herrero, J., Gómez-Navarro, C. & Zamora, F. Intrinsic electrical conductivity of nanostructured metal-organic polymer chains, *Nat. Commun.* **4**, 1709 (2013).

(39) Li, M.-Y., [...].

[...]

Minor typographical errors have been corrected, the methods expanded and the presentation of figures improved where necessary.

Thank you very much for kind comments.

Response for the Reviewer 2

I am now satisfied with the revised manuscript for publication in Nature Communications.
Please include the IR region for polarized reflectance spectra in SI.

Thank you very much for kind comments. The polarized reflectivity spectra together with IR region were added as Supplementary Fig. 4 as follows.

(Results in the original manuscript)

Figure 3 and Supplementary Fig. 3 show polarized reflectivity spectra of the 1–2 heterostructure.

→

(Results in the revised manuscript)

Figure 3 and Supplementary Figs. 3 and 4 show polarized reflectivity spectra of the 1–2 heterostructure.

Supplementary Fig. 4 | The polarized reflectivity spectra for parallel (||: red lines) and perpendicular (⊥: blue lines) toward the chain direction of the (a) **1**, (b) heterojunction, and (c) **2** regions in the **1–2** heterostructure. The insets show optical images with apertures (UV-vis-NIR region: $\phi = 100 \mu\text{m}$; IR region: $100 \mu\text{m} \times 100 \mu\text{m}$). Scale bars $100 \mu\text{m}$ (for **a–c**). Source data are provided as a Source Data file.

Response for the Reviewer 3

I recommend publishing the revised manuscript as a Nature Communication. The authors have addressed my concerns with the original manuscript.

Thank you very much for kind comments.